# Upgrade of the NA61/SHINE Facility beyond 2020 for an Expanded Physics Programme

**Dag Larsen** 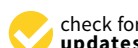 **and on behalf of the NA61/SHINE collaboration**

Institute of Physics, Jagiellonian University, 31-007 Kraków, Poland; dag.larsen@cern.ch

**Abstract:** The NA61/SHINE experiment studies hadron production in hadron-hadron, hadron-nucleus and nucleus-nucleus collisions. The physics programme includes the study of the onset of deconfinement and search for the critical point as well as reference measurements for neutrino and cosmic ray experiments. For strong interactions, future plans are to extend the programme of study of the onset of deconfinement by measurements of open-charm and possibly other short-lived, exotic particle production in nucleus-nucleus collisions. This new programme is planned to start after 2020 and requires upgrades to the present NA61/SHINE detector setup. Besides the construction of a large acceptance silicon detector, a 10-fold increase of the event recording rate is foreseen, which will necessitate a general upgrade of most detectors.

**Keywords:** NA61/SHINE; upgrade; open charm; silicon detector; time projection chamber; CERN long shutdown 2

## 1. Introduction

The NA61/SHINE experiment [1] is preparing for a 10-fold increase of the read-out rate to satisfy the requirements for the proposed physics programme [2] after CERN long shutdown 2 (beyond 2020), in particular related to the open-charm measurement with the vertex detector (VD). This requires an upgrade of most detectors and other sub-systems of the NA61/SHINE experiment.

The NA61/SHINE charm program is a natural extension of the previous studies of the phase transition to the quark-gluon plasma. It addresses the question of the validity and the limits of statistical and dynamical models of high energy collisions in the new domain of quark mass, $m_c \approx 1300$ MeV $\gg T_C \approx 150$ MeV. Among many questions which might be answered by the new NA61/SHINE program, there are three that primarily motivate it:

- What is the mechanism of charm production?
- How does the onset of deconfinement impact charm production?
- How does the formation of quark gluon plasma impact $J/\psi$ production?

To answer these questions, knowledge is needed on the mean number of charm–anti-charm quark pairs $\langle c\bar{c} \rangle$ produced in the full phase space of heavy ion collisions. Such data do not exist yet and NA61/SHINE aims to provide them within the coming years. Figure 1 presents a compilation of predictions by dynamical and statistical models on $\langle c\bar{c} \rangle$ produced in central Pb + Pb collisions at $158A$ GeV/c. These predictions are obtained from:

- The hadron string dynamics (HSD) model [3]—a perturbative quantum chromo-dynamics (pQCD)-inspired extrapolation of p + p data.
- A pQCD-inspired model [4,5]—calculation based on model assumptions and nucleon parton density functions only.

- The hadron resonance gas model (HRG) [6]—a calculation of equilibrium yields of charm hadrons assuming parameters of a hadron resonance gas fitted to mean multiplicities of light hadrons.
- The statistical quark coalescence model [6]—a statistical distribution of $c$ and $\bar{c}$ quarks between hadrons. The mean number $\langle c\bar{c} \rangle$ of charm pairs is calculated using the measured $\langle J/\psi \rangle$ multiplicity [7] and the probability of a single $c\bar{c}$ pair hadronising into a $J/\psi$ calculated within the model.
- The dynamical quark coalescence model [8]— quark coalescence as a microscopic hadronisation mechanism of deconfined matter. The mean number $\langle c\bar{c} \rangle$ of charm pairs is calculated using the measured $\langle J/\psi \rangle$ multiplicity [7] and the probability of a single $c\bar{c}$ pair hadronising into a $J/\psi$ calculated within the model.
- The statistical model of the early stage (SMES) [9]—the mean number of charm quarks is calculated assuming an equilibrium quark–gluon-plasma (QGP) at the early stage of the collision.

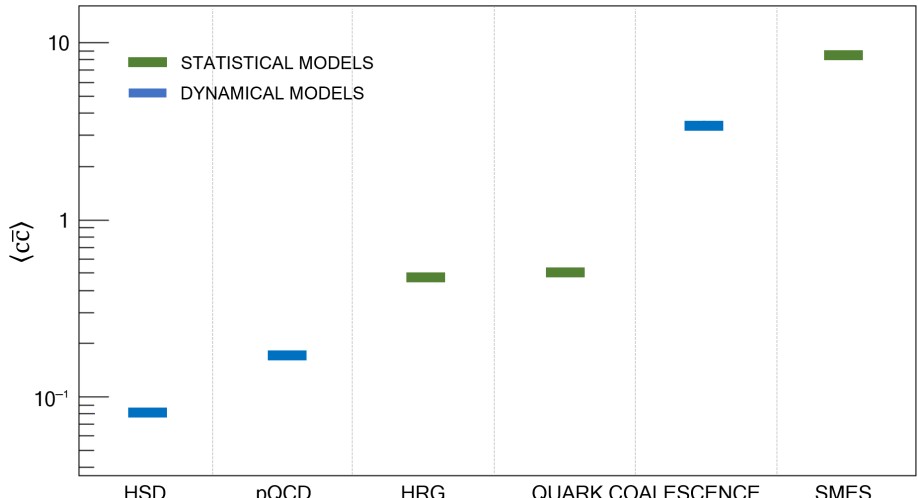

**Figure 1.** Mean multiplicity of charm quark pairs produced in the full phase space in central Pb + Pb collisions at 158$A$ GeV/c calculated with dynamical models (blue bars): hadron–string-dynamics (HSD) [3], pQCD–inspired [4,5], and dynamical quark coalescence [8], as well as statistical models (green bars): HRG [6], statistical quark coalescence [6], and SMES [9].

The predictions of the models on $\langle c\bar{c} \rangle$ differ by about two orders of magnitude. Therefore, obtaining precise data on $\langle c\bar{c} \rangle$ is expected to allow to narrow the spectrum of viable theoretical models and thus learn about the charm quark and hadron production mechanisms. The production of charm is expected to be different in confined and deconfined matter. This is caused by differences of the charm carriers in these phases. In confined matter the lightest charm carriers are D mesons, whereas in deconfined matter the carriers are charm quarks. Production of a D$\overline{\text{D}}$ pair ($2m_D = 3.7$ GeV) requires an energy about 1 GeV higher than production of a $c\bar{c}$ pair ($2m_c = 2.6$ GeV). The effective number of degrees of freedom of charm hadrons and charm quarks is similar [10]. Thus, more abundant charm production is expected in deconfined than in confined matter. Consequently, in analogy to strangeness [9,11], a change of collision energy dependence of $\langle c\bar{c} \rangle$ may be a signal of onset of deconfinement.

Figures 2 and 3 present the collision energy dependence of charm production in central Pb + Pb collisions at 150$A$ GeV/c predicted by two very different models: the statistical model of the early stage [10] and a pQCD-inspired model [12], respectively. Figure 2 shows the energy dependence of $\langle c\bar{c} \rangle$ predicted by the statistical model of the early stage.

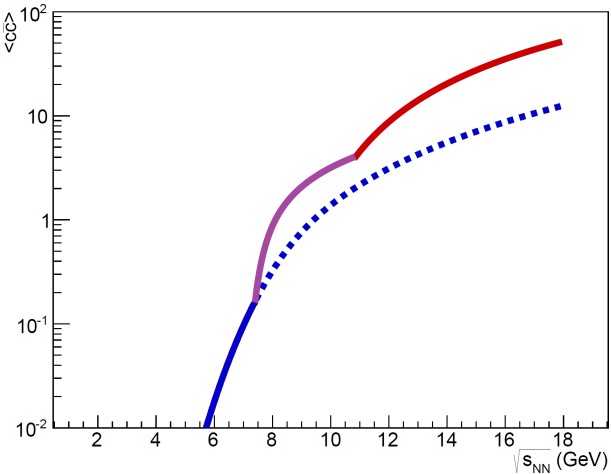

**Figure 2.** Energy dependence of $\langle c\bar{c} \rangle$ in central Pb + Pb collisions calculated within the SMES model [10]. The blue line corresponds to confined, the purple line to mixed phase, and the red line to deconfined matter. The dashed line presents the prediction without a phase transition.

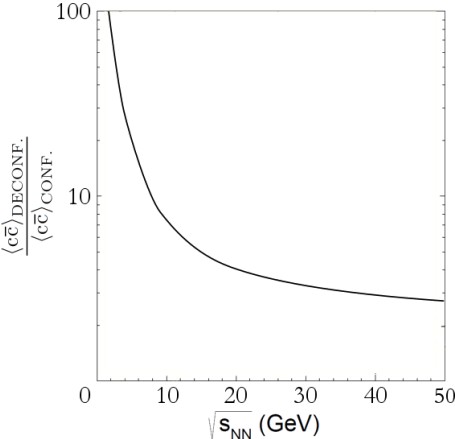

**Figure 3.** Energy dependence of the ratio of $\langle c\bar{c} \rangle$ in deconfined and confined matter in central Pb + Pb collisions calculated within the pQCD-inspired model of Ref. [12].

According to this model, when crossing the phase transition energy range ($\sqrt{s_{\mathrm{NN}}} = 7 - 11$ GeV), an enhancement of $\langle c\bar{c} \rangle$ production should be observed. At $150A$ GeV/c ($\sqrt{s_{\mathrm{NN}}} = 16.7$ GeV ) an enhancement by a factor of about 4 is expected. Figure 3 shows the ratio of mean multiplicity of $c\bar{c}$ pairs in deconfined and confined matter calculated with the pQCD-inspired model of Reference [12]. Both numerator and denominator were evaluated at the same collision energy. At $150A$ GeV/c ($\sqrt{s_{\mathrm{NN}}} = 16.7$ GeV ) an enhancement by a factor of about 3 is predicted. Accurate experimental results will allow to test these predictions. Suppression of the production of $J/\psi$ mesons in central Pb + Pb collisions at $158A$ GeV/c was an important argument for the CERN announcement of the discovery of a new state of matter [13]. Within the Matsui-Satz model [14] the suppression is attributed to the formation of the QGP.

Figure 4 presents two scenarios of charmonium production. In the first case (Figure 4 left), a produced $c\bar{c}$ pair hadronises in vacuum—this corresponds to a p + p interaction. Open charm and charmonia are produced in vacuum with a certain probability, at high collision energies typically 10% of $c\bar{c}$ pairs form charmonia and 90% appear in open charm hadrons.

The second scenario is illustrated in Figure 4 right. Here the $c\bar{c}$ pair forms a pre-charmonium state in the quark gluon plasma. Due to the colour screening, which may lead to disintegration of this state, the probability of charmonium production is suppressed in favour of open charm production.

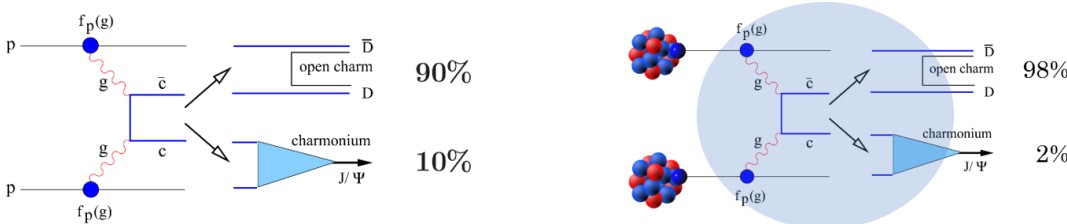

**Figure 4.** Sketch of the charmonium production mechanism and its relation to $c\bar{c}$ production in p + p interactions left and central heavy ion collisions right [15].

The probability of a $c\bar{c}$ pair hadronising to J/$\psi$ is defined as: $P(c\bar{c} \rightarrow J/\psi) \equiv \frac{\langle J/\psi \rangle}{\langle c\bar{c} \rangle} \equiv \frac{\sigma_{J/\psi}}{\sigma_{c\bar{c}}}$. To be able to calculate this probability, one needs data on both J/$\psi$ and $c\bar{c}$ yields in full phase space. At the CERN SPS precise $\langle J/\psi \rangle$ data was provided by the NA38 [16], NA50 [7], and NA60 [17] experiments, while $\langle c\bar{c} \rangle$ data is not available at the CERN SPS energies. The problem of the lack of $\langle c\bar{c} \rangle$ data was worked around [7,14] by assuming that the mean multiplicity of $c\bar{c}$ pairs is proportional to the mean multiplicity of Drell-Yan pairs: $\langle c\bar{c} \rangle \sim \langle DY \rangle$.

Figure 5 shows the result from the NA50 experiment [7] that was interpreted as evidence for QGP creation in central Pb + Pb collisions at 158$A$ GeV/c based on this assumption. However, the assumption $\langle c\bar{c} \rangle \sim \langle DY \rangle$ may be incorrect due to many effects, such as shadowing or parton energy loss [18]. This clearly shows the need for precise data on $\langle c\bar{c} \rangle$ in centrality selected Pb + Pb collisions at 150$A$ GeV/c.

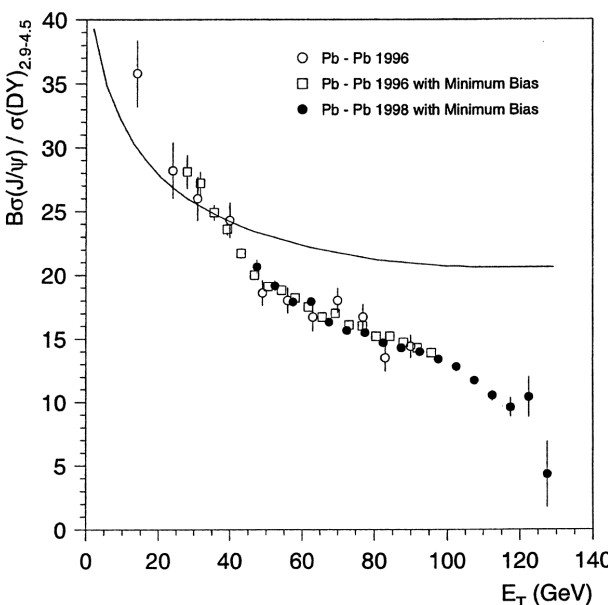

**Figure 5.** The ratio of $\sigma_{J/\psi}/\sigma_{DY}$ as a function of transverse energy (a measure of collision violence or centrality) in Pb + Pb collisions at 158$A$ GeV/c measured by NA50. The curve represents the J/$\psi$ suppression due to ordinary nuclear absorption [7].

## 2. Vertex Detector (VD) Upgrades

The upgrade of the existing Small Acceptance Vertex Detector (SAVD) detector [19] aims to adapt this detector to the requirements of data taking with a 1 kHz trigger rate, and to increase its geometrical acceptance. Both measures will yield significant ($\sim$30$\times$) increase of statistics of the reconstructed decays of charm hadrons.

To fulfil its task, the novel vertex detector (VD) will have to provide a rate capability, which exceeds the one of the SAVD by one order of magnitude. The required data rate is found to exceed the capabilities of the successful MIMOSA-26AHR [20] sensors used so far in the SAVD. Moreover, the related increase of the radiation doses has to be considered. The requirements on the sensors can

be estimated by scaling the corresponding numbers derived for the SAVD [2] to the novel running scenario. This is done assuming that the VD will operate at a true collision rate of 5 kHz, and at a duty cycle of 0.15. The results for the most exposed point of the VD and an operation time of 40 days are shown in Table 1. No safety margin was considered. One finds the radiation load is dominated by the beam halo of direct beam ions, which varies substantially depending on the quality of the beam tuning. The table provides two numbers based on the scaling of the initial assumptions as described in the reference and based on a measurement of the beam halo carried out during the 2017 Xe + La at 150$A$ GeV/c run. One observes that optimising the beam for a low beam halo yielded a significant lowering of the radiation load. Despite remaining moderate, the ionising radiation doses still exceed the radiation tolerance of the previously used MIMOSA-26AHR sensors.

**Table 1.** Radiation doses for the most exposed point of the VD and a run of 40 days with 150$A$ GeV/c Pb + Pb. For the radiation dose generated by the beam halo, numbers scaled from Ref. [2] (scaled) and numbers relying on measurements of the beam halo performed during the 2017 Xe + La run at 150$A$ GeV/c (measured) are shown. The beam halo of the Xe–beam was reduced by means of careful beam tuning. The related numbers are considered as most representative for future experiments.

| Radiation Source | Ionising | Non-Ionising |
|---|---|---|
| | (krad) | ($10^{12}$ $n_{eq}$/cm$^2$) |
| Direct particles | 35 | 1.3 |
| Delta electrons | 40 | Negligible |
| Beam halo (scaled) | 1200 | 2.0 |
| Beam halo (measured) | 200 | 0.3 |
| Sum requirements | 275–1275 | 3.3 |
| ALPIDE | $> 500$ | 17.0 |

The novel detector will reuse the mechanics and infrastructure of the SAVD. A photograph of the SAVD just before its installation on the beam for the test measurement in 2016 is shown in Figure 6. One can see vertically oriented carbon fibre ladders with MIMOSA-26 sensors installed in their centres as well as the Pb target of 1 mm thickness located about 50 mm upstream from the first SAVD station. The carbon fibre ladders are exactly the same as those used in the Inner Barrel of the new ALICE inner tracking system (ITS).

Despite of the good experience with the MIMOSA-26AHR sensors in the construction and operation of the SAVD, the VD cannot be built using these sensors: in order to cope with the 10-fold increase in beam intensity and interaction rate a better time resolution is required (by a factor of 10). The ALPIDE sensor and the detector concept developed for the new ALICE ITS is considered as the best candidate for the VD in 2022. In December 2016 one ITS Inner Barrel stave with 9 ALPIDE chips, the green vertical structure in Figure 6, was already successfully operated in NA61/SHINE during five days of the test with Pb + Pb collisions at 150$A$ GeV/c. Discussions concerning further collaboration and technology transfer already started.

In the spirit of the above considerations the upgraded VD will rely on the carbon fibre support structures developed for the ALICE ITS. Instead of the older MIMOSA-26AHR, they will host the modern ALPIDE CMOS Pixel Sensors [21]. A comparison of the features of both sensors is given in Table 2.

The novel sensors come with a powerful $\sim 1$ Gbps data interface and a time resolution of 10 μs, which is by more than one order of magnitude faster than the average time between two collisions. This fact and the capability of ALPIDE to use external trigger information for data reduction ensures the rate capability required for the VD. As shown in Table 1, the sensor also matches the requirements in terms of radiation dose. To estimate its tolerance to direct ion hits in terms of single event errors (SEE), the sensor was operated for several days in a direct Xe-beam at the SPS in 2017 and no crucial

incident was observed. This suggests that the chip is not particularly vulnerable and that no dedicated detector safety system for the case of beam displacement is required. Note that the tolerance of ALPIDE to ionising radiation does likely exceed the 500 krad guaranteed so far. This is a subject of ongoing research.

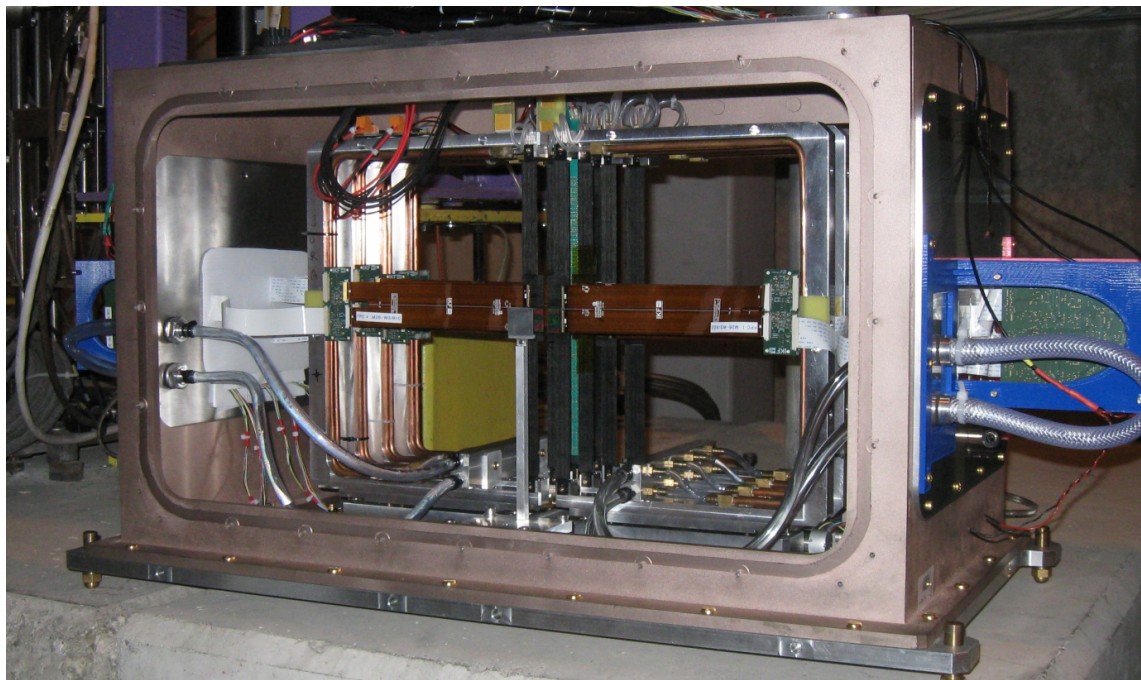

**Figure 6.** The SAVD used by NA61/SHINE during the data taking in 2016 and 2017.

**Table 2.** Comparison of basic parameters of MIMOSA and ALPIDE sensors.

|  | MIMOSA-26AHR | ALPIDE |
|---|---|---|
| Sensor thickness (μm) | 50 | 50 |
| Spatial resolution (μm) | 3.5 | 5 |
| Dimensions (mm$^2$) | $10.6 \times 21.2$ | $13.8 \times 30$ |
| Power density (mW/cm$^2$) | 250 | 40 |
| Time resolution (μs) | 115.2 | 10 |
| Detection efficiency (%) | >99 | >99 |
| Dark hit occupancy | $\lesssim 10^{-4}$ | $\lesssim 10^{-6}$ |

As the fibre supports were initially designed for ALPIDE, the carbon fibre plates required for adapting them mechanically to MIMOSA-26AHR become obsolete, the material budget is reduced slightly by 0.1% $X_0$. Moreover, accounting for the very low power consumption of ALPIDE, it is considered not to use the active cooling foreseen in the support structures. The absence of coolant in the structures would once more reduce the material budget of the VD as compared to the SAVD. The obsolete front-end cards and readout electronics of the SAVD will be replaced as well.

The planned extension of the VD aims to increase its geometrical acceptance from 33% (SAVD) to 70% of the tracks also detected in the TPCs of NA61/SHINE. This number holds for a Pb + Pb collision system at 150$A$ GeV/c. To reach this goal, it is necessary to extend the detector from 10 ladders for the present SAVD to 16 ladders, which will hold 46 ALPIDE sensors with a total active surface of 190 cm$^2$. The future sensor configuration is displayed in Figure 7 and the related Geant4 model used for the performance simulations may be found in Figure 8. Note that the upgrade will require only minor modifications in the mechanical design, as the SAVD was already designed for compatibility with ALPIDE sensors and the related ladders.

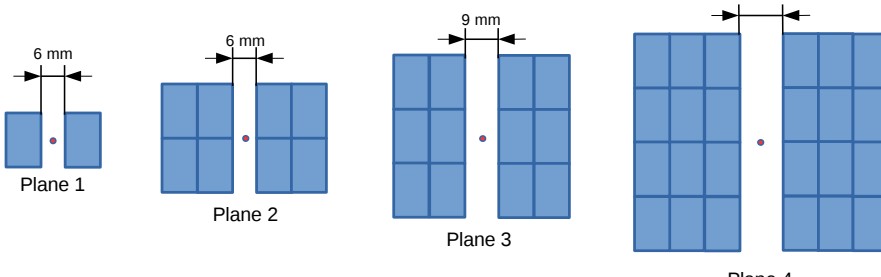

**Figure 7.** Schematic view of the VD layers based on ALPIDE sensors. From left to right: the first layer with two sensors, the second layer with 8 sensors, the third layer with 12 sensors and the fourth layer with 24 sensors. The total active area of the VD sensors is 190 cm$^2$.

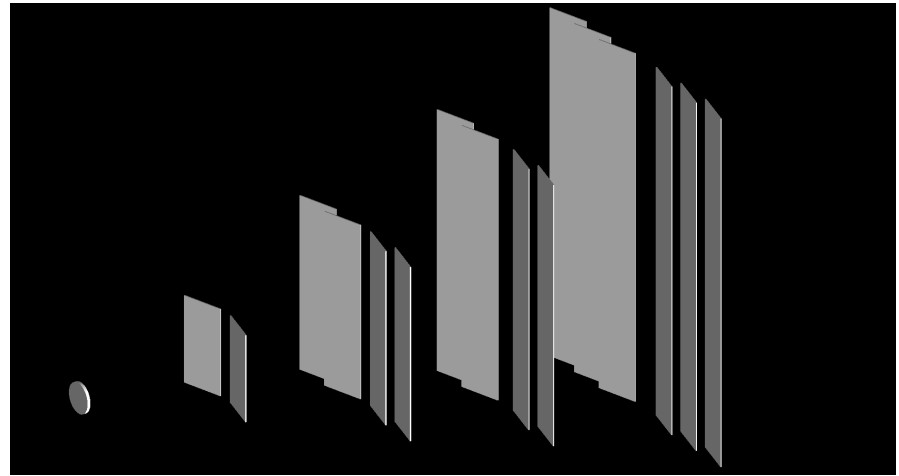

**Figure 8.** The Geant4 visualisation of the VD detector geometry described in Figure 7.

## 3. Time Projection Chamber (TPC) Upgrades

The increase of the readout speed of the time projection chambers (TPCs) is an essential and integral part of the upgrade of NA61/SHINE for the charm physics program. The goal is to reach a readout rate of 1 kHz. To achieve this NA61/SHINE can profit from the fact that the ALICE experiment at the LHC is replacing their wire chamber readout of the TPC. This implies also the exchange of the complete readout electronics chain. NA61/SHINE has signed a memorandum of understanding which defines the transfer of part of the ALICE TPC readout electronics to the collaboration.

The design of the NA61/SHINE TPC is very similar to the design of the ALICE TPC. Therefore it is not surprising that the readout electronics shows strong similarities in its key parameters listed in Table 3. The most relevant difference is the higher digitisation rate which in the end allows a readout rate up to a factor 10 higher than presently possible in NA61/SHINE. In addition the dynamic range is considerably higher due to the 10 bit ADCs. Also the higher sensitivity, the lower noise level and the doubling of the number of time bins should be mentioned here. For the transport of the data from the front-end electronics to the DAQ system ALICE developed a second generation readout control Unit (RCU) called RCU2. Its segmentation into four 40 bit wide readout busses where each readout bus can connect up to 8 front-end cards (FECs) and a 300 MByte/s optical link provides sufficient bandwidth.

**Table 3.** Comparison between key parameters of the NA61/SHINE and the ALICE front-end electronics.

|  |  | NA61/SHINE | ALICE |
|---|---|---|---|
| Signal polarity |  | positive | positive |
| Signal width (FWHM) | ns | 180 | 190 |
| Dynamic range |  | 120:1 | 900:1 |
| MIP S:N ratio |  | 14:1 | 14/20/18:1 |
| Noise | e | 1100 | <1000 |
| ADC number of bits |  | 8 | 10 |
| Number of time slices |  | 512 | 1000 |
| Power consumption | mW/ch | 51 | 35 |
| Sampling rate | MHz | 5, 10 | 5, 10 |
| Readout frequency | MHz | 0.1 | 5, 10 |
| Integrated non-linearity | % | <2 | 0.2 |

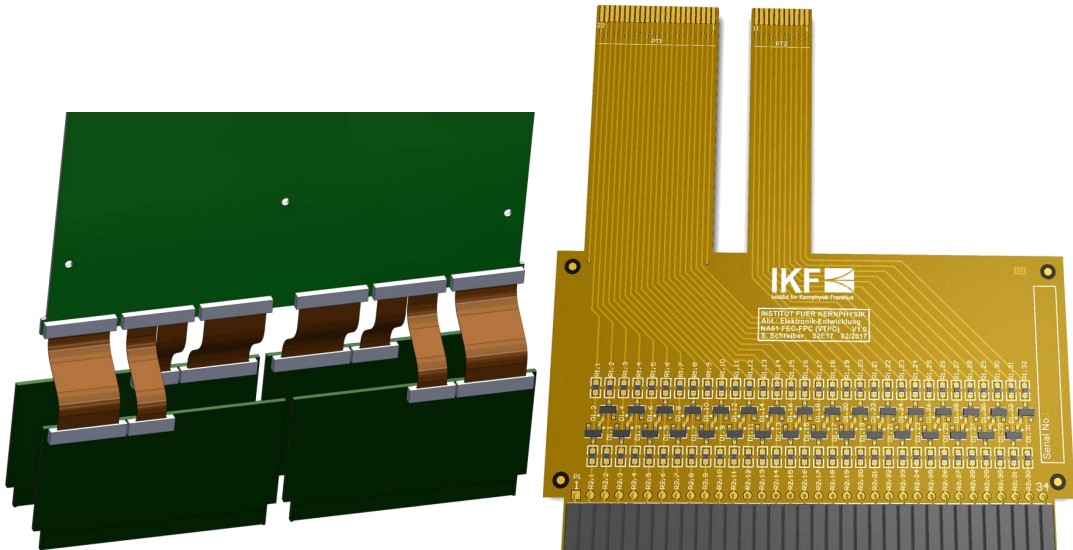

**Figure 9.** (**Left**) simulation showing an ALICE FEC with adapter cards and flexible kapton cables. (**Right**) details of the design of a kapton adapter cable including the protection circuit.

In the following an overview of the various steps necessary to replace the existing NA61/SHINE readout by the ALICE components is presented.

- Development, construction and tests of prototype input adapter cables: Due to the big differences in readout connector topology and the large difference in the number of channels per front-end card (32 vs. 128), a careful study of the design of adapter cables connecting the existing 32 channel NA61/SHINE connectors to the six 22/21 channel input cables of the ALICE FECs was performed. A possible scenario for the connection of the FECs to the NA61/SHINE wire chambers is shown in Figure 9 left panel. Furthermore, the operation of ALICE has shown a weakness in the input protection circuit (incorporated into the design of the pre-amplifier/shaper CMOS chip) of the ALICE FECs. Therefore an additional protection circuit based on surface mounted (SMD) components will be incorporated into the adapter cables. Based on a three-dimensional (3-D) model, prototype adapter boards are being developed and built, see Figure 9 right panel. The very first test of the protection network and the effect of the increased length of the adapter cables on the noise performance of the FECs will be tested in an existing test stand. Further tests are then foreseen on-detector in NA61/SHINE with a single FEC connected to the upstream corner of a MTPC chamber, using a special adapter/test setup from ALICE.
- Design of the mechanical support of the FECs: For the MTPCs no major problems are expected as sufficient space is available. However, the confined space in the vertex TPCs requires a special

arrangement of the FECs (mounting under an angle). This is presently studied together with the design of the adapter cards on the output side of the FECs.

- Design and implementation of the FEC cooling: One option is to cool the FECs following the present NA61/SHINE scheme and even use the existing cooling plates. Another option is the use of fans to remove the heat. The choice will be made once the mounting schemes have been developed and the space constraints are better known.

- Production and tests of interface boards for the connection of the FEC output to the flexible busses: Due to the different topologies the ALICE FECs cannot be read out via a rigid bus as in ALICE. Instead a more flexible readout using flat cables has to be used. Such a cable readout has already been developed for the PHOS detector in ALICE using the same FECs and RCUs as the TPC, see Figure 10. This know-how can be directly applied to the future NA61/SHINE readout. Nevertheless, the production and tests of the various adapter boards (to FEC and to RCUs) require a considerable effort, see Figures 11 and 12.

- Development of read-out and DAQ for the new electronics: The new readout requires a new DAQ system. A new modular system partly based on the ALICE HLT design is under investigation. The development and test of a new TPC DAQ system will be done in parallel to the development of the hardware components described above.

- Laboratory tests of the new readout chain: Tests of the readout of several FECs in the lab using the full read-out chain (FECs, flexible cables, small adapter boards, RCU2s) will be performed. These tests will be repeated with FECs connected to the upstream corner of one MTPC chamber with beam in order to see real track signals.

- Design and implementation of a new Low Voltage system: The new readout system requires power supplies, bus bars and cables for the distribution of the low voltage (LV). It is foreseen to follow the design of the ALICE TPC LV system. For the distribution of the LV inside the chambers a system with bus bars running on one side of the chambers will be used, very similar to ALICE. The connection between bus bars and FECs is then done by short patch cables.

- Development and implementation of new detector control system (DCS): The new readout system requires a considerable extension of the existing NA61/SHINE DCS system to make full use of the information supplied by the FECs and the RCUs. It will follow the software design developed for ALICE.

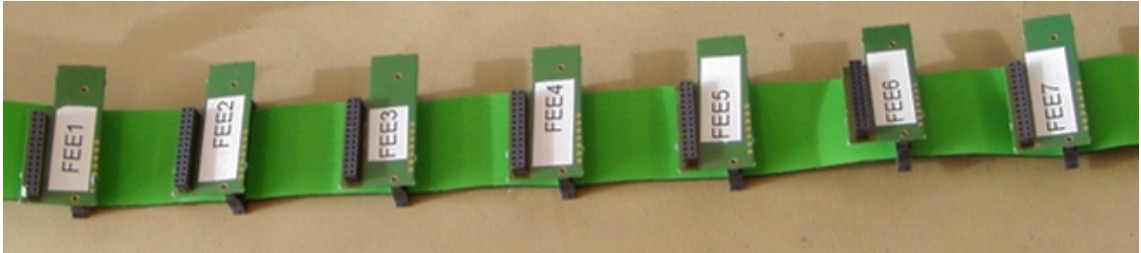

**Figure 10.** One of the three flexible readout bus cables with adapters.

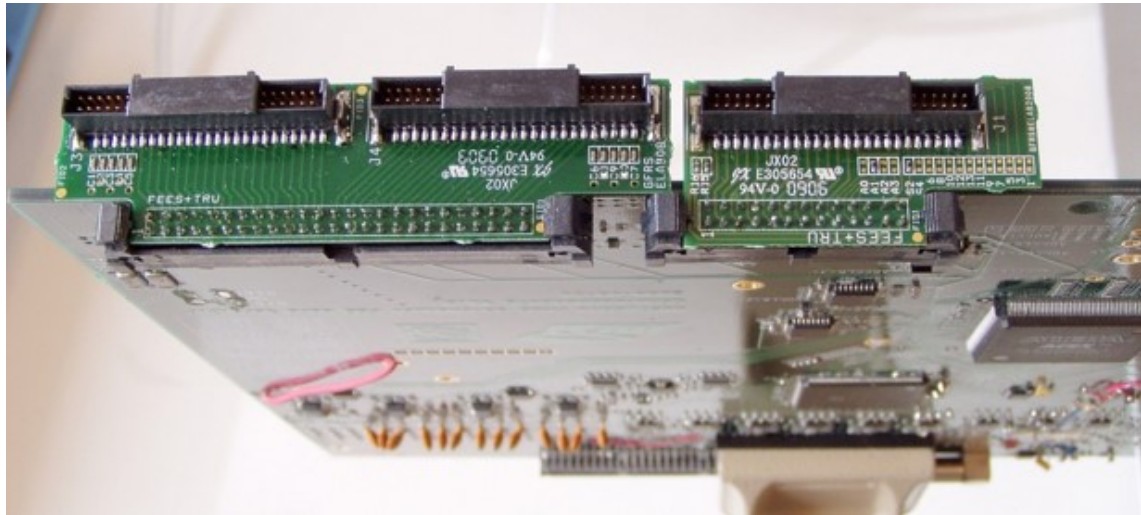

**Figure 11.** Front-end card with its two adapter cards to connect to the flexible readout busses.

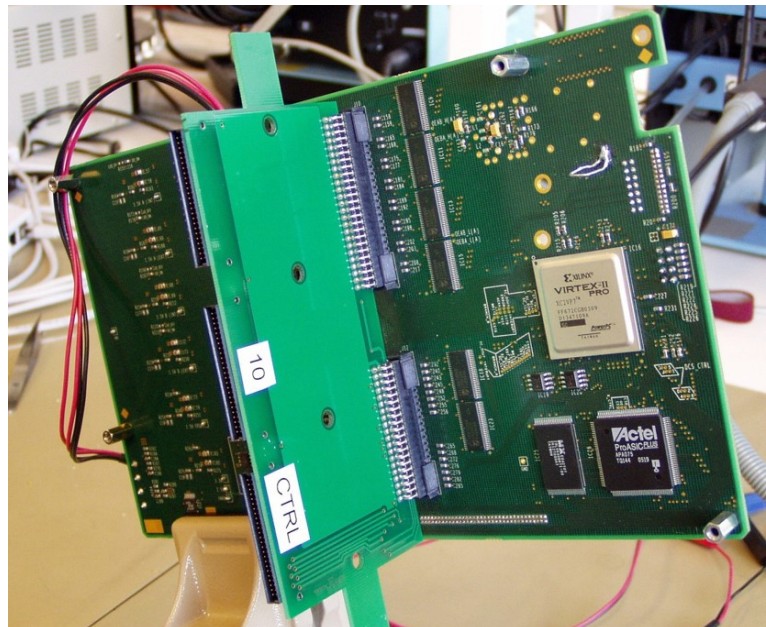

**Figure 12.** Read-out control unit (RCU) with the adapter card plugged on.

## 4. Projectile Spectator Detector (PSD) Upgrade

A forward hadron calorimeter, the PSD, measures forward energy (mostly from projectile spectators) and allows to reconstruct the event plane independently from the tracking in the TPCs. In addition, a fast analogue signal from the PSD is used to select at the trigger level events based on the measured forward energy. The NA61/SHINE physics program beyond 2020 requires a tenfold increase of the beam and trigger rates. This necessitates an upgrade of the PSD.

The present PSD consists of 16 central small modules with transverse sizes of $10 \times 10$ cm$^2$ and 28 outer large modules with transverse sizes $20 \times 20$ cm$^2$ (Figure 13 left). The length (depth) of the modules is 5.6 interaction lengths. The present PSD has no beam hole in the centre. A small additional module is installed in front of the centre of the PSD to improve the energy reconstruction for heavy fragments (Figure 13 right). The present PSD has a good response linearity and energy resolution $\frac{\sigma_E}{E} = \sqrt{\left(\frac{0.65}{\sqrt{E}}\right)^2 + 0.033^2 + \left(\frac{2.7}{E}\right)^2}$ according to measurements with proton beams in the energy range

20–158 *A* GeV/c (see Figure 14). Results are shown for the case when the proton beam hits one of the PSD central modules.

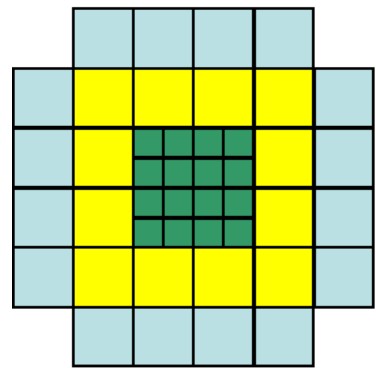 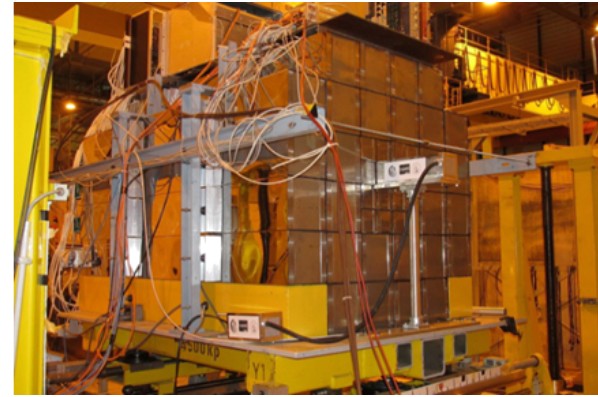

**Figure 13.** (**Left**) schematic front view of the present PSD of NA61/SHINE. (**Right**) Photo of the PSD placed on the beam line downstream of the NA61/SHINE detector. An additional small module (1.8 interaction length) is installed in the front of the PSD.

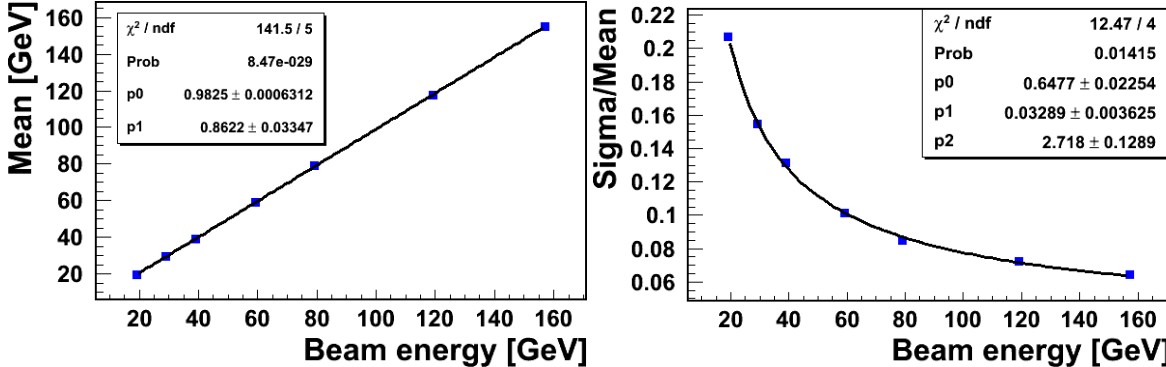

**Figure 14.** Response linearity (**left**) and energy resolution (**right**) of the PSD.

Figure 15 shows the radiation dose and neutron fluence simulated with the FLUKA code for the present PSD. The Pb beam rate was assumed as $5 \times 10^4$ ions per second. The accumulated radiation dose during one month of data taking exceeds $10^3$ Gy for the central part of the present PSD. Light transparency of the scintillator tiles degrades significantly above this dose. Moreover, the neutron fluence is of the order of $10^{12}$ n/cm$^2$ near the beam axis behind the calorimeter. This would cause degradation of the performance of the MPPC photo-detectors (increase of dark current, drop of gain, etc.). placed at the rear side of the calorimeter as well as of the commercial FPGAs used in the present readout electronics and also situated at the rear side of each module. Thus, the radiation hardness problems will lead to the deterioration of reliability and response of the calorimeter.

The increase of the lead ion beam intensity by more than one order of magnitude (up to $10^5$ ions per second) requires upgrades of the radiation hardness and protection as well as of the readout rate of the PSD. In the present PSD only 16 small central modules use fast Hamamatsu MPPC photodetectors. Rather old MAPD-3A photodiodes with slow pixel recovery time are used in the other 28 large modules. Figure 16 shows the dependence of the signal amplitude on the proton beam rate for one of the PSD sections. There is no reduction of the MPPC amplitude at a beam rate of $10^5$ protons per second. The present readout electronics is based on 33 MSPS ADCs. Neither the slow photo-diodes nor the readout electronics are suitable for the higher beam intensity planned for NA61/SHINE beyond 2020.

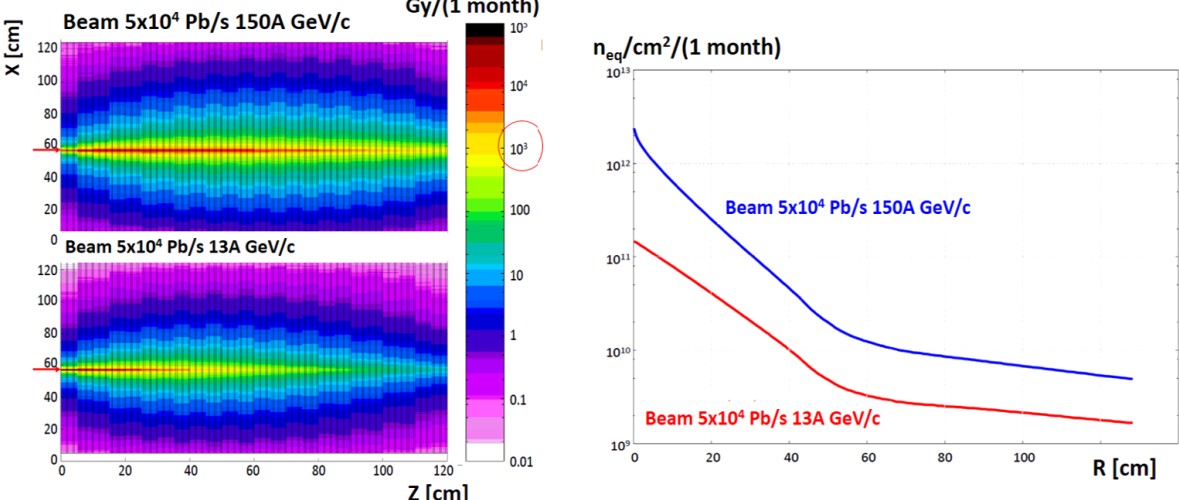

**Figure 15.** (**Left**) radiation dose distribution along the PSD for Pb ions of 150$A$ GeV/c and 13$A$ GeV/c. (**Right**) neutron fluence as function of distance from the beam axis at the rear side of the PSD for the same beam momenta.

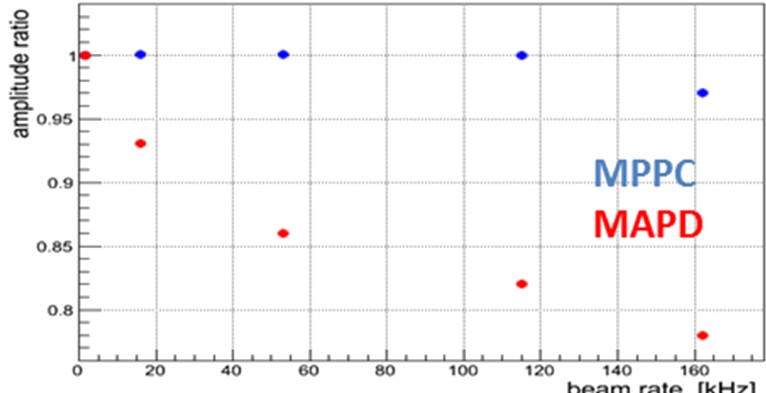

**Figure 16.** MPPC and MAPD signal amplitudes as a function of proton beam rate in one of the PSD sections.

The increase of the Pb beam rate by more than one order of magnitude will lead to a radiation alarm in the experimental area of NA61/SHINE, because the present PSD serves as an active beam dump. Therefore, the PSD must be protected by additional concrete shielding. This is practically impossible for the present calorimeter because it is placed on a movable platform with large transverse size which is used to change the position of the PSD during data taking runs. To solve the problems mentioned above, it is proposed to use two calorimeters, the Main (M-PSD) and a Forward (F-PSD), see Figure 17, instead of the present PSD. The M-PSD would be based on the present PSD with the 16 small central modules replaced by four new central modules with transverse sizes 20 × 20 cm$^2$ and with truncated edges forming a beam hole of 60 mm diameter at the centre. In addition, 8 cm thick boron polyethylene blocks placed at the rear side of each of these modules will reduce the neutron fluence in the front of the MPPCs. The F-PSD is an additional small calorimeter placed at a distance of 4.6 m downstream of the M-PSD, see Figure 18. It consists of 9 modules with transverse sizes of 20 × 20 cm$^2$. All F-PSD modules will have 5.6 $\lambda_{\text{int}}$ interaction lengths, the same as in the M-PSD, except for a longer central module of 7.8 $\lambda_{\text{int}}$. As for the M-PSD, 8 cm thick boron polyethylene blocks will be placed at the rear side of each module. According to simulations, the two calorimeter setup will decrease the hadron shower leakage for Pb + Pb interactions at 150$A$ GeV/c from 11% for the present PSD to 4%.

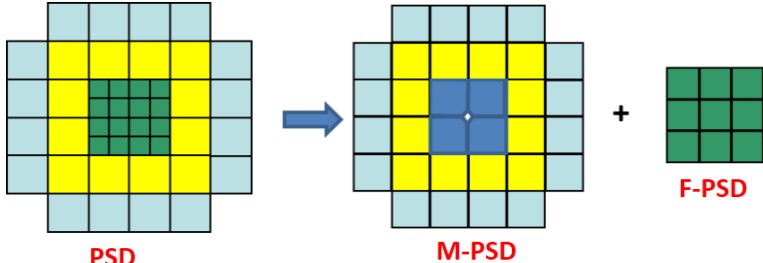

**Figure 17.** Schematic front view of the current PSD and the proposed new M-PSD and F-PSD calorimeters.

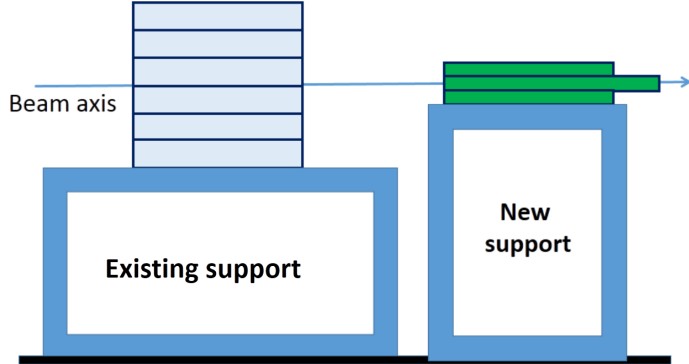

**Figure 18.** Horizontal cut of the proposed two calorimeter setup for NA61/SHINE.

The distribution of the radiation dose simulated with the FLUKA code for Pb + Pb collisions at $150A$ GeV/c are shown in Figure 19 for the M-PSD and the F-PSD. The radiation dose and neutron fluence for the M-PSD are at an acceptable level even in the central modules. The expected radiation dose in the central module of the F-PSD is large, leading to loss of transparency of the scintillator tiles. But because the F-PSD will measure mainly heavy fragments producing a large amount of light, the attenuation can be compensated by increasing the MPPC bias voltages. The neutron fluences for the MPPCs in the F-PSD are at an acceptable level. Clearly, permanent monitoring is necessary for the F-PSD during data taking.

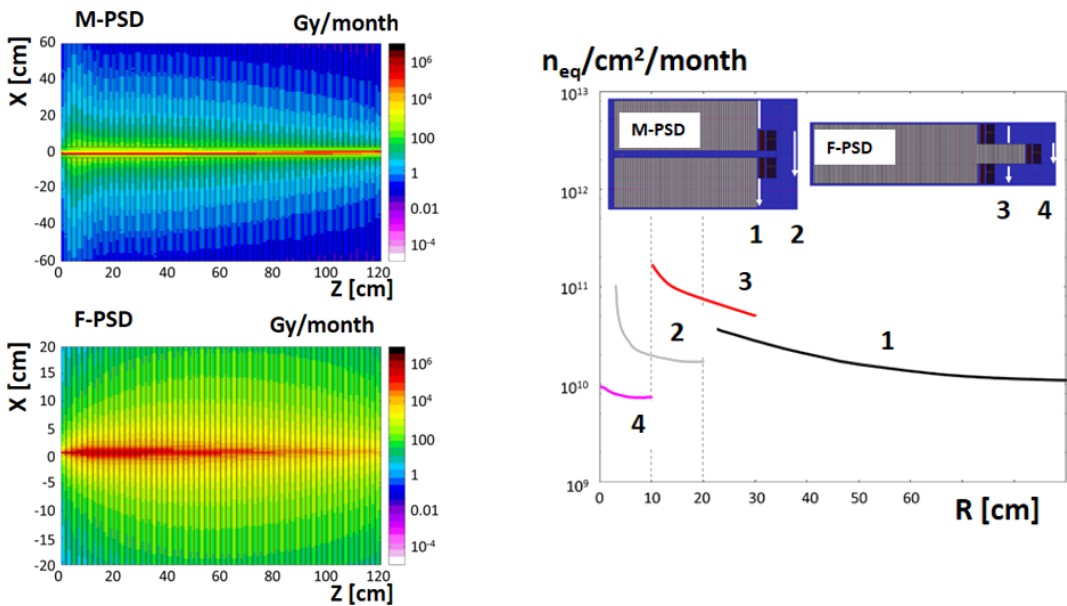

**Figure 19.** Radiation dose distributions along the M-PSD and the F-PSD (**left**) and neutron fluence distributions as a function of distance from the beam axis at the rear sides of the M-PSD and F-PSD (**right**) for the two calorimeter setup.

It is also important to take into account the activation of the M-PSD and F-PSD due to the high beam rate. Permitted activation for access is 0.5 μSv/h. According to simulations, the M-PSD activation will decrease to the permitted level already after one day without the beam. The activation of the F-PSD is significantly higher and decreases to the permitted level of activation only 6 months after stop of the beam. Additional concrete shielding of the F-PSD will be required for radiation protection.

The precision of the reaction plane determination for Pb + Pb collisions at 150*A* GeV/c with the present PSD and with the two calorimeter setup is shown in Figure 20 left and right, respectively. One concludes that the precision of the reaction plane determination for semi-peripheral collisions remains almost unchanged.

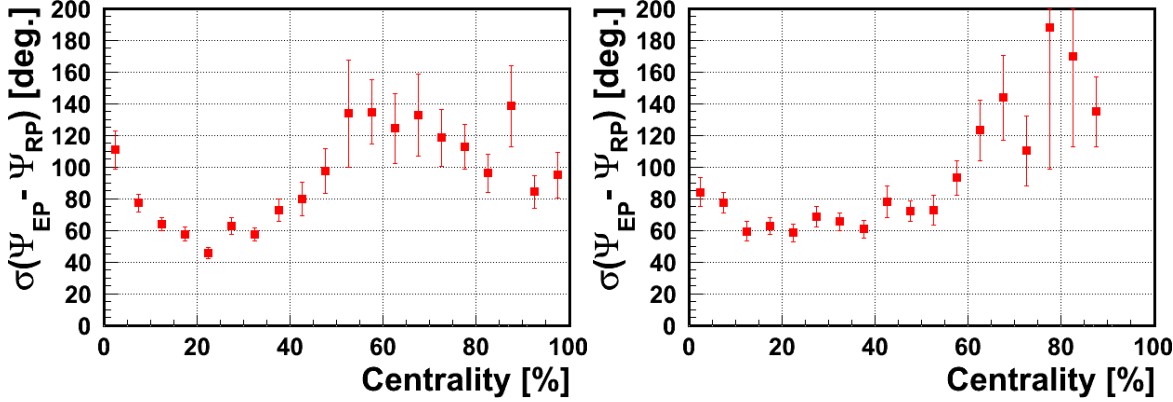

**Figure 20.** Reaction plane resolution as a function of centrality in Pb + Pb collisions at 150*A* GeV/C determined with the present PSD (**left**) and with the M-PSD and F-PSD setup (**right**).

## 5. Trigger and DAQ Upgrade

The key motivation for the readout system upgrade is the need to increase the event flow rate from 80 Hz to 1 kHz. Furthermore, the evolution of the NA61/SHINE physics program requires adding new sub-detectors to the Data AcQuisition system (DAQ) easily without in-depth knowledge of the system. Also excluding sub-detectors from the DAQ system, which are not required for a particular run, would be beneficial to limit the event size. The current DAQ system is already at the performance limit regarding the number of detectors as well as the bandwidth. Adaptation of the system would require substantial modifications. Consequently, it was decided to design a new system which meets the future requirements of NA61/SHINE. The most fundamental requirements regarding the new DAQ system are:

- Speed—1 kHz readout frequency
- Robustness—extended self diagnostic and adaptation algorithms of the DAQ core.
- Facilitate Control—shifter friendly interface to monitor and control data taking with algorithms detecting pre-failing states.
- Use of commercial off-the-shelf (COTS) components—profit from industry progress and competitive prices.
- Homogeneous Core—data from all subsystems treated in the same way.
- Inhomogeneous Nodes—each sub-detector readout system can be freely chosen by the sub-detector group.
- Extendable—adding new detectors in plug-and-play manner. Self subscription (Nodes) and self adaptation (Core).
- Transparency—detector developers have well defined interface to pack, send and unpack data. The DAQ details will be hidden from sub-detector developers.

Currently, the raw event size is about 50 MB which after compression (zero-suppression algorithm) is reduced to about 1.5–5 MB depending on event multiplicity [22]. Adding new detectors will increase

the event size and consequently the cost of hardware and data storage. To keep the overall cost within a reasonable range, an event size limit has to be introduced. A maximum size of 20 MB per event should be sufficient for the NA61/SHINE program beyond 2020. In order to achieve such high readout speed, a network has to have a bandwidth of 20 GB/s: 1 kHz × 20 MB = 20 GB/s = 160 Gb/s. For the time being the 100 Gb Ethernet is being evaluated as the technology for the DAQ core. The attractive price, low number of links as well as flexibility due to high throughput makes 100 GbE a promising technology. Additionally, in order to use the full potential of this technology two techniques are explored: Remote Direct Memory Access (RDMA) and network package aggregation. The RDMA reduces CPU involvement in data transmission by bypassing the Linux kernel network stack. By so doing, the number of buffer copy operations between Open Systems Interconnection (OSI) layers is decreased and consequently the CPU time is saved. The latter technique—network package aggregation—prevents throughput degradation due to protocol overhead when dealing with very small packages. Therefore it improves performance greatly when small packages are aggregated into one big buffer and sent afterwards.

The general schematic of the new readout system is depicted by Figure 21. The *Detector Layer* is the starting point of data flow. This layer consists of front-end electronics, which sends data to the next layer – Readout Layer. The technology used to transport data between those layers as well as everything inside the Detector Layer are not part of the DAQ project. Thus sub-detector groups can freely choose the technologies they want to use. This design describes the data flow starting with the Readout Layer.

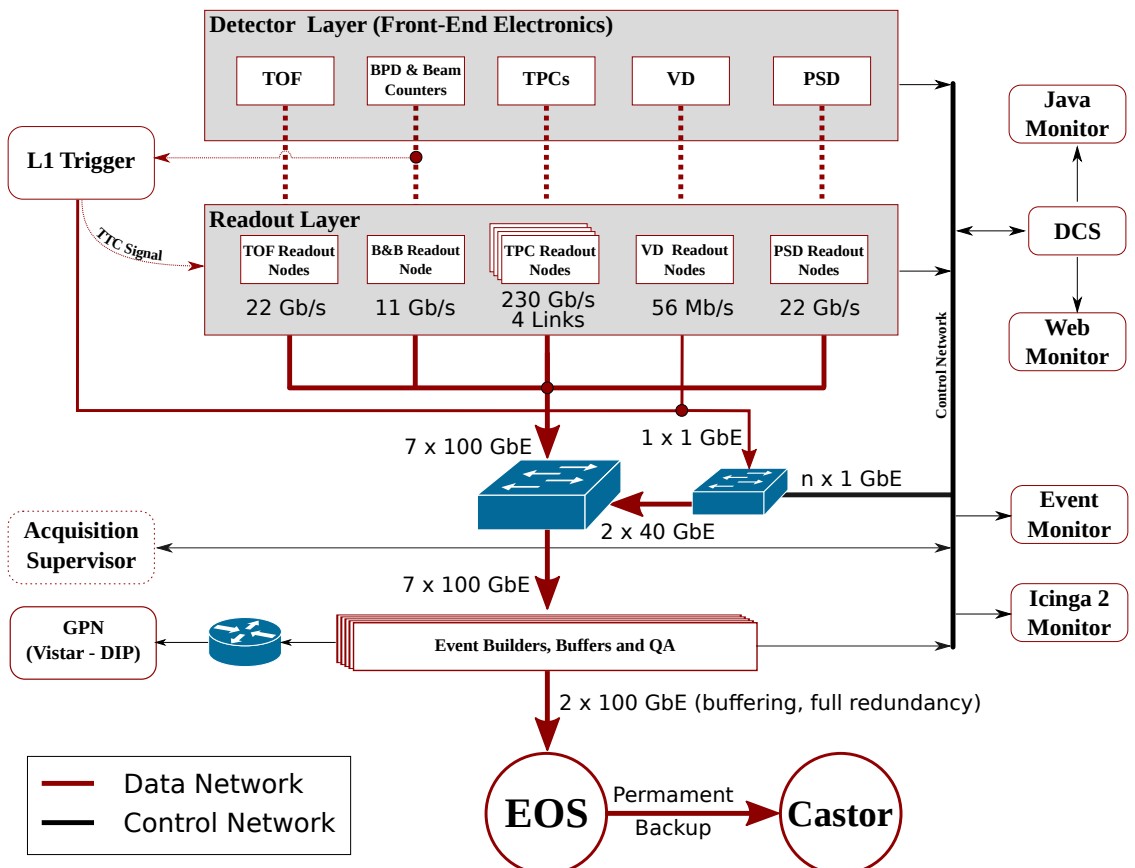

**Figure 21.** Overview of the planned NA61/SHINE data acquisition system.

The *Readout Layer* consists of nodes, which are sub-event builders. In other words, a node collects data from the Detector Layer, builds sub-events and sends them to an Event Builder. Nodes are required to utilise x86-64 architecture and Ethernet technology for connection with Event Builders



(through the Switching Network). The connection can be chosen as 1 GbE or 100 GbE depending on the amount of data produced by the sub-detector. The new DAQ system will provide a library to facilitate sending sub-events and communication with the Acquisition Supervisor as well as monitoring facility. Furthermore, the library will also implement a software RAM-based buffer. Therefore, nodes should have 8 GB of RAM at minimum.

For the time being, the design foresees only a level 1 trigger (*L1 Trigger*). In order to improve the time response of the trigger, it has to take input signals directly from the front-end electronics of the beam counters. Signals from other sub-detectors can be used as additional trigger input signals, if needed.

The *Acquisition Supervisor* is an integral and central part of the DAQ system and, at the same time, the most crucial one. It is meant to perform the following functions:

- Sending/Updating LookUp Table (LUT) of Event Builder IP addresses.
- Sending control commands such as *START*, *STOP* etc.
- Collecting diagnostic data e.g. Buffer occupancy, CPU load etc.
- Performing damage control tasks, e.g. abandoning non-responsive event builders.
- Providing the DAQ User Interface (UI) for expert and non-expert users.

The *Event Builders* receive sub-events from all nodes and form a final event. The event is stored in an internal RAM-based buffer until a chunk (set of events) is formed (1 GB size) which is then sent to the final storage. The *Storage* will be delivered by CERN such as CASTOR, EOS etc. However, due to frequent network bandwidth problems between the experiment and the CERN storage services, an intermediate storage, which can buffer the data taking of up to three days, is under consideration. The *Vistar* [23] webtools will be used to publish the most crucial information about data taking so that experts can easily monitor the situation. In addition, the *Event Monitor* and the *DCS* will be integrated with the DAQ system.

In order to reduce the required data storage requirements, additionally to the trigger system, a partial online reconstruction (clustering) and an off-line filtration will be used. The online clustering is already implemented in the TPC readout electronics of the ALICE readout system. The first test with one TPC sector is planned for the summer of 2018. The prototype of the off-line filtration system has already been implemented successfully on the OpenStack platform. Tests proved that the data can be filtered out with a low latency of around one hour after data had been collected. Furthermore, construction of the DAQ testbed has already started in November 2017, using more than 30 multi-core dual CPU machines. Before 2020, the testbed will be used for prototyping of the new data acquisition system. Afterwards, it is meant to be the DAQ replica, used for developing additional features and further improvement of the existing ones. Consequently, all developments will be done on the testbed, independently from the production DAQ, thus without a risk of jeopardising data taking.

## 6. Conclusions & Outlook

The NA61/SHINE collaboration has performed a detailed study of the requirements for the various detectors and sub-systems resulting from the necessary 10-fold increase of the data readout rate. It was shown that fulfilling these requirements is feasible, and detailed plans for the upgrades of the different systems have been developed and are now being implemented to ensure successful operation of the upgraded experiment beyond 2020.

**Author Contributions:** The author presents a large collaborative effort over a long period of time. Individual contributions to the effort can not trivially be identified.

**Funding:** The author and the work related to the SAVD hardware development and experimental data taking was supported by the Polish National Center for Science grants 2014/15/B/ST2/02537 and 2015/18/M/ST2/00125. The overall current NA61/SHINE experiment facility is funded by numerous grants over a long period of time. The grant applications for upgrades are currently being reviewed.

**Conflicts of Interest:** The author declares no conflict of interest.

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
