# Peer review of "Upgrade of the NA61/SHINE Facility beyond 2020 for an Expanded Physics Programme"

_universe, doi:10.3390/universe5010024_

Round 1
Reviewer 1 Report
As the script introduces a very technical upgrade, which should be checked and evaluated. Because the upgrade sounds very convincing and needed. The improvement of 10 times faster read-out and the integrability with the ALICE detector adds to the argumentation favoring a strong recommendation for publication in MDPI.
Author Response
Thank you very much for your comments.
Reviewer 2 Report
In this manuscript, the author discusses the detailed upgrade strategy of the NA61/SHINE experiment, which is planned to start after 2020. The plan includes the extension of the vertex detector, the increase of the readout speed and event recording rate, and the upgrade of the data acquisition system. Overall the presentation quality is good, and the discussions are thorough. However, the physics case for this upgrade is not clear in the current version of the manuscript, as it is only briefly mentioned in the abstract. I think it would be better that the author considers expanding the introduction and conclusion before publication, by explaining what is the physics goal of the project, i.e., is the main purpose of the experiment the study of the onset of deconfinement, or does it also aim for other discoveries?
Author Response
Thank you for your comments. I have expanded the introduction to include the physics case. (I was initially planing to include it, but decided against it as the technical part became rather extensive. But as a second thought, this probably was a mistake.)